# Fabrication and Properties of a Bio-Based Biodegradable Thermoplastic Polyurethane Elastomer

**DOI:** 10.3390/polym11071121

**Published:** 2019-07-02

**Authors:** Zhaoshan Wang, Jieqiong Yan, Tongyao Wang, Yingying Zai, Liyan Qiu, Qingguo Wang

**Affiliations:** 1Key Laboratory of Rubber-Plastics of Ministry of Education, Qingdao University of Science and Technology, Qingdao 266042, China; 2Shandong Provincial Key Laboratory of Rubber-Plastics, Qingdao 266042, China

**Keywords:** polyurethane, thermoplastic, polycondensation, biodegradability, bio-based

## Abstract

Using the melt polycondensation of five bio-based aliphatic monomers (succinic acid, sebacic acid, fumaric acid, 1,3-propanediol, and 1,4-butanediol), we first synthesized the more flexible and biodegradable polyester diols (BPD) with an average molecular weight of 3825. Then, the BPD was polymerized with excessive 4,4′-diphenylmethane diisocyanate (MDI). Finally, the molecular chain extender of 1,4-butanediol (BDO) was used to fabricate the biodegradable thermoplastic polyurethane elastomer (BTPU), comprising the soft segment of BPD and the hard segment polymerized by MDI and BDO. Atomic force microscope (AFM) images showed the two-phase structure of the BTPU. The tensile strength of the BTPU containing 60% BPD was about 30 MPa and elongation at break of the BTPU was over 800%. Notably, the BTPU had superior biodegradability in lipase solution and the biodegradation weight loss ratio of the BTPU containing 80% BPD reached 36.7% within 14 days in the lipase solution.

## 1. Introduction

Thermoplastic polyurethane (TPU) is an important polymer that combines the softness and elasticity of rubber with the easy processability of thermoplastics [1]. Simultaneously, TPU has many advantages [2,3], such as simple processing technology, recyclability, superior mechanical and thermal properties [4,5], excellent wear resistance [6,7], and biocompatibility [8,9]. TPU is irreplaceable in various fields, including automotive, screens, roller systems films, medical, sports products, aerospace and electronics, and electronic applications [10,11,12,13].

Unfortunately, traditional preparation methods of TPU rely on fossil resources, which undoubtedly exacerbates the depletion of fossil resources [14,15]. Moreover, most TPUs are not biodegradable [16]. Thereby, improving the biodegradability of TPU and reducing the use of nonrenewable resources have become urgent and significant subjects in the field of polymer materials.

Biomass residues, such as corncob [17], cocopeat [18] and sugarcane bagasse [19], are important resources for preparing some bio-based monomers. Bio-based monomers or polymers have many advantages, including low pollution, low cost, and superior biodegradability [20,21,22]. Therefore, efficiently developing and utilizing biomass residues will play positive roles in solving energy and ecological problems [23,24,25,26].

In recent years, the application of biomass residues in preparing bio-based biodegradable polymer materials has become an important focus in the polymer field [27,28], in which biodegradable thermoplastic polyurethane (BTPU) has also attracted a lot of attention. Sit-Foon Cheng et al. [29], using palm oil-based polyester polyol, synthesized water-blown porous biodegradable polyurethane foams with a mass loss ratio in the range from 10 to 40% in two weeks under enzymatic conditions. Meanwhile, the tensile strength of the polyurethane foams was only in the range from 0.05 to 0.3 MPa and the elongation at break of the polyurethane foams was in the range from 52 to 227%. Kucharczyk, P. et al. [30] synthesized a Polylactic acid (PLA)-based polyesterurethane with an elongation at break of about 3% and good biodegradability. Gokhan Acik et al. [31] synthesized a series of soybean oil-based biodegradable rigid polyurethane films (PU-Fs), which had a glass transition temperature (*T*_g_) in the range from 55 to 62 °C. After 12-week enzymatic experiments, the mass of the rigid PU-Fs was lost by about 40 to 50%.

Being different from TPUs, which cannot have good biodegradability and good elasticity simultaneously, this paper—using bio-based aliphatic monomers [32,33] and polyester esterification reaction and chain extension technologies—designed and prepared a novel bio-based BTPU, which demonstrates both good biodegradability and good elasticity. BTPU contains a novel soft segment of aliphatic polyester and a hard segment. Notably, compared to some soft segments with 1–3 polyester repeat units in some traditional TPUs [29,30,31], the novel soft segment of the novel BTPU comprises six kinds of aliphatic polyester repeat units and has good biodegradability and flexibility. Also, the use of fumaric acid is explored to synthesize the novel soft segment because the carbon-carbon double bond in the main molecular chain has the effect of improving the molecular chain relaxation rates [34], leading to a more flexible soft segment. This paper also studied the effects of the molecular structure, molecular weight and soft segment ratio on the biodegradability, mechanical property and thermal property of BTPU. This work can provide new research insights for developing environmentally friendly and functional biodegradable polymer materials.

## 2. Experimental Section

### 2.1. Materials

1,3-propanediol (PDO) with a molecular weight of 76.10 gmol^−1^ and a purity of 99% was supplied by Hunan Hainabaichuan Bioengineering Co., Ltd., Yiyang, China. 1,4-butanediol (BDO) with a molecular weight of 90.12 gmol^−1^ and purity of no less than 99% was supplied by Shouguang Jinyu Chemical Co., Ltd., Shouguang, China. Succinic acid (SU) with a molecular weight of 118.09 gmol^−1^ and purity of no less than 99% was supplied by Shandong LanDian Biological Technology Co., Ltd., Shouguang, China. Sebacic acid (SA) with a molecular weight of 202.25 gmol^−1^ and purity of no less than 98.5% was supplied by Inner Mongolia Weiyu Biological Technology Co., Ltd., Tongliao, China. Fumaric acid (FA) with a molecular weight of 116.07 gmol^−1^ and purity of no less than 99.5% was supplied by Anhui Sealong Biotechnology Co., Ltd., Bengbu, China. 4,4-diphenylmethane diisocyanate (MDI), with a molecular weight of 250.26 gmol^−1^ was supplied by Wanhua Chemical Group Co., Ltd., Yantai, China. Para-toluene sulfonic acid with a molecular weight of 172.20 gmol^−1^ and purity of no less than 99.0% was supplied by Sinopharm Chemical Reagent Co., Ltd., Shanghai, China. Chloroform with a molecular weight of 119.38 gmol^−1^ and purity of no less than 99.0% was supplied by Yantai Sanhe Chemical Reagent Co., Ltd., Yantai, China. Anhydrous methanol, with a molecular weight of 32.04 gmol^−1^ and purity of no less than 99.5% was supplied by Tianjin Damao Chemical Reagent Factory, Tianjin, China. Lipase, 20000 U, was supplied by Henan Xinyu Food Additives Co., Ltd., Zhengzhou, China. Mixed phosphate (pH buffer), was supplied by Shanghai Leici Chuangyi Instrument Co., Ltd., Shanghai, China. Antioxidant-168 (AT-168) and antioxidant-1010 (AT-1010) were supplied by Beijing HWRK Chemical Co., Ltd., Beijing, China.

### 2.2. Experimental Instruments

The ATR-IR analyzer (VERTEX 70) was bought from Bruker Corp., Leipzig, Germany. The nuclear magnetic resonance (NMR) spectrometer, AVANCE, was from Bruker Corp., Leipzig, Germany. The gel permeation chromatograph (GPC), HLC-8320, was manufactured by Tosoh Corp., Tokyo, Japan. The differential scanning calorimeter (DSC), 204F1, was supplied by Netzsch Corp., Selb, Germany. The thermogravimetric analyzer (TGA), 209F1, was manufactured by German Netzsch Corp., Selb, Germany. The atomic force microscope (AFM), Multimode8, was from Bruker Corp., Leipzig, Germany. The electronic tensile machine, AI-7000, was bought from Taiwan Gotech Testing Machines Inc, Taichung, Taiwan.

### 2.3. Experimental Methods

#### 2.3.1. Synthesis of Biodegradable Polyester Diols (BPD)

The monomers of PDO, BDO, SA, SU, and FA were placed into a four-necked round-bottom flask (500 mL), which was equipped with magnetic stirring, thermometer, and nitrogen gas entrance. The mole ratio of PDO/BDO/SA/SU/FA was 5.9/5.9/5.6/1.4/3, in which the mole ratio of alcohol to acid was 1.18:1 and FA accounted for 30% of the moles of the dibasic acid. Under the protection of a nitrogen atmosphere, the reaction mixture was slowly heated to 190 °C until the monomers completely melted. Then, the esterification reaction was retained for 2 h and the water produced was removed with the help of nitrogen gas. The catalyst (para-toluene sulfonic acid with a weight ratio of 0.25%) and antioxidants (AT-168 and AT-1010 with a weight ratio of 0.1%) were subsequently added into the above polyester prepolymer, followed by decompressing the pressure of the reactor below 200 Pa and increasing the temperature of polyester prepolymer from 190 °C to 220 °C. The polycondensation reaction was not terminated until the acid value of the reaction system was below 1 mg KOH/g, then the biodegradable polyester diol (BPD) was fabricated.

#### 2.3.2. Fabrication of BTPU

As listed in Table 1, BPD and MDI with a certain weight ratio were added into a four-necked flask (250 mL) equipped with magnetic stirring and a thermometer. Under vacuum conditions, the reaction was continued at 80 °C for 120 min. Then, a certain quantity of BDO was slowly added into the reactor as a chain extender under a fast stirring speed. After 12 h of aging in a vacuum-drying oven at 100 °C to 120 °C, BTPU-1, BTPU-2, and BTPU-3 were obtained. Figure 1 shows the schematic illustration for the formation of BTPU from the monomers.

### 2.4. Characterization and Testing Methods

#### 2.4.1. Analysis by Fourier Transform Infrared (FTIR) Spectrometer

The BPD and BTPU synthesized above were dissolved in chloroform and precipitated with cooling methanol. After drying, the purified BPD and BTPU were obtained for testing.

The purified BPD and BTPU samples were tested using the attenuated total reflectance mode, with a resolution of 4 cm^−1^ and scanning wave numbers ranging from 4000 cm^−1^ to 600 cm^−1^.

#### 2.4.2. Analysis by Proton Nuclear Magnetic Resonance (^1^H-NMR) Spectrometer

Purified BPD and BTPU were tested at a frequency of 300 Hz and a temperature of 25 °C using tetramethylsilane as the internal standard and deuterated chloroform as the solvent.

#### 2.4.3. Analysis by Gel Permeation Chromatograph

The synthesized BPD and BTPU with a concentration of 0.2–0.5 wt% were characterized at 35 °C for the detector and column using tetrahydrofuran (THF) as the solvent, with a flow rate at 0.35 mL/min. Shodex-polystyrene was used as the standard for calibration.

#### 2.4.4. Atomic Force Microscope (AFM) Observation

The micromorphological structure of the synthesized BTPU was observed by AFM in the tapping mode. The scope of the scanner was 10 μm × 10 μm. The spinning speed of spin coater was 1000 r/min and the time was 60 s. The method of making the sample was as follows. The BTPU was dissolved in the solvent THF to make a dilute solution with a concentration of about 0.02 wt%. Then, the solution was dropped onto the newly exfoliated mica sheet and the film was formed by spinning. The solvent was evaporated for a period of time before testing and the dried spin-coated sample film was then tested by AFM.

#### 2.4.5. Mechanical Property Test

The tensile properties were tested according to ISO 37-2017 with a stretching speed of 200 mm/min. The thickness of the narrow portion was 1.0 mm ± 0.1 mm for dumbbell specimens. The test length was 10 ± 0.5 mm.

The hardness was tested according to ISO 48-4:2018. For the determination of hardness using a type A durameter, the thickness of the test piece was at least 6 mm. The other dimensions of the test pieces were sufficient to permit measurement of at least 12 mm away from any edge for the type A durameter. The standard test time was 15 s for thermoplastic rubber.

#### 2.4.6. Thermal Property Test

The synthesized BPD and BTPU were measured with a differential scanning calorimeter. The measurement conditions are as follows. With a blanket of nitrogen air for protection, the dried sample was heated from room temperature to 250 °C and maintained for 5 min. Then, the sample was cooled down to −90 °C and heated again to 250 °C. The heating and cooling rates were both 10 °C/min.

The synthesized BPD and BTPU were tested with the thermogravimetric analyzer at the following conditions: The temperature was heated from room temperature to 900 °C under a nitrogen atmosphere at a heating rate of 20 °C/min and a flow rate of 50 mL/min.

#### 2.4.7. Biodegradation Performance Test

The lipase degradation method was as follows [35,36]. Approximately 0.5 g of the dried BTPU sample, weighed as *W*_0_, was placed into a tube. Subsequently, approximately 0.01 g of lipase was added in the test tube and the total mass (sample, tube, and lipase) was denoted as *W*_1_. An appropriate amount of mixed phosphate buffer solution (about 8 mL) with a pH value of 6.8 was added into the sampling tube. This solution was then placed into a thermostatic water bath at a constant temperature of 37 °C. The tube was taken out every 48 h. After centrifugal separation and discarding the solution, the residual solid in the tube was dried in a vacuum-drying oven to a constant weight. The final total mass was denoted as *W*_2_. Finally, the mass loss rate was calculated using Equation (1):(1)$$\mathrm{Mass}\;\mathrm{loss}\;\mathrm{rate}\;=\;\left[\left(W_{1}-W_{2}\right)/W_{0}\right]\times 100\%$$

Approximately 0.01 g of lipase and a specific amount of mixed phosphate buffer solution with a pH of 6.8 were weighed and added into the sampling tube to repeat the test, which ensured that similar biodegradation conditions were supplied for the biodegradation tests of the remaining BTPUs in the following biodegradation tests.

## 3. Results and Discussion

### 3.1. Structural Characterization of BPD and BTPU

#### 3.1.1. FTIR Analysis of BPD and BTPU

Figure 2 shows the infrared spectra of the purified BPD and BTPU. In the infrared spectrum of BPD, the stretching vibration absorption peak of hydroxyl (-OH) occurred at 3553 cm^−1^. The characteristic peak of the carbonyl group (C=O) and the stretching vibration absorption peak of the ether linkage (C-O-C) appeared at 1723 and 1153 cm^−1^, respectively. Moreover, they were both strong. The above three peaks indicated that the molecular structure of the synthetic BPD contained a large number of hydroxyls (-OH) and ester groups (-COO-). Being different with the spectrum of BPD, the -OH characteristic peak at 3553 cm^−1^ disappeared in the infrared spectrum of BTPU, and the -NCO absorption peak disappeared at 2270 cm^−1^, which indicated that the -OH of BPD completely reacted with the isocyanate group (-NCO) of MDI. An absorption peak occurred at 1597 cm^−1^, which corresponded to the carbon-carbon double bond (C=C) stretching vibration on the aromatic ring. The result proved the existence of the aromatic ring structures of MDI in the molecular chain of BTPU. Moreover, the N-H stretching vibration absorption peak occurred at 3332 cm^−1^, meanwhile, the N-H bending vibration absorption peak occurred at 1531 cm^−1^, which suggested that the new N-H groups were generated by the reaction between -NCO and -OH. The appearance of two absorption peaks at 1722 (C=O) and 1153 cm^−1^ (C-O-C) indicated that structures of the amido bond (-NHCO-) existed in the molecular chain of BTPU.

#### 3.1.2. H-NMR Analysis of BPD and BTPU

Figure 3 shows the molecular structural formulas of BPD and BTPU. Figure 4 presents the ^1^H-NMR spectra of BPD and BTPU. In the ^1^H-NMR spectra of BPD, the peak j (δ = 3.43 ppm) was the resonance absorption peak of H in the hydroxyl-terminated absorption peak. The peaks at 6.82(i) ppm could be attributed to the carbon-carbon double bond of FA. Moreover, no hydroxyl-terminated absorption peak of H was found in the ^1^H-NMR spectra of BTPU. However, a resonance absorption peak of H in -NH- appeared at position m (δ = 7.31 ppm) and a resonance absorption peak of H on the carbon directly connected to the -(C=O)-O-CH_2_- appeared at position p (δ = 4.10 ppm) in the spectra of BTPU. At the same time, the peak n (δ = 7.09 ppm) corresponded to the resonance absorption of H in the benzene ring and the peak l (δ = 3.80 ppm), which corresponded to the resonance absorption peak of H in -CH_2_- between two benzene rings appeared. The other absorption peaks of BTPU were coincident with the peaks of BPD or had some small deviations, which showed that the molecular structure of BTPU was generated by the reaction among BPD, MDI, and BDO.

#### 3.1.3. Molecular Weight and Molecular Weight Distribution of BTPU

Table 2 shows the molecular weight of several BTPUs with different BPD contents. When the content of BPD (soft segment) in BTPU increased from 60 wt to 80 wt%, the molecular weight of BTPU increased and the molecular weight distribution became wider. This could be explained, as with the increasing of the BPD content, the number of hydroxyl groups in BPD increases and the corresponding isocyanate groups in MDI decrease. This results in the ratio of the isocyanate groups in MDI to the hydroxyl groups in BPD being close to 1, which led to the increase of the molecular weight of the prepolymer. Then, the average molecular weight and the weight average molecular weight of BTPU are both increased by using BDO to extend the molecular chains.

### 3.2. Micromorphological Structure of BTPU

Figure 5a–c indicates the 2-D atomic force microscopic images of BTPU-2. Obvious bright lumpy bulges could be found in the phase image and the inerratic bulges, corresponding to the phase image, could be found in the height image and amplitude error image. Thereinto, the bright convex part was the hard segment of BTPU due to its high surface energy and modulus, and the dark concave portion was the soft segment of BPD. The interface between the hard and soft segments was clear, the phase domain was large and the phase separation was apparent. Figure 5a–c shows the 3-D images of BTPU-2. The dark part distributed in the trough of the wave was the soft segment of BTPU, which was distributed as a continuous phase. Meanwhile, the hard segment was the light-colored convex part of the image, which was the dispersed phase. Figure 5 accurately reflects the microphase separation of BTPU-2, from which we could see that the hard segments were more evenly distributed in the soft segments in BTPU-2.

### 3.3. Mechanical Properties of BTPU

Table 3 lists some mechanical properties of the three kinds of BTPUs. When the soft segment content in BTPU increased from 60 wt to 80 wt%, the tensile strength of BTPU decreased from 30.2 MPa to 13.5 MPa, the elongation at break increased from 802 to 1405%, and the hardness gradually decreased from 87 HA to 60 HA.

The reasons for these changes are as follows. The hard segment of BTPU is prone to oriented crystallization, which forms the crystalline region. The BPD in an amorphous state surrounds this region. Thus, the crystalline region forms the physical crosslinking point of the BTPU material. Under the action of an external force, the tensile strength of the crystalline region is higher than that of the amorphous region and the elongation at break is smaller than that of the amorphous zone. Therefore, the relative hard segment content decreased and the orientational crystallization ability of the segment weakened with the increase of the BPD content in BTPU, which resulted in a decrease in the tensile strength of BTPU. Meanwhile, the density of the physical crosslinking point network consisted of crystallizing points decreased with the decrease in the proportion of the crystalline phase (or the number of crystalline dots). This condition weakened the binding force to the surrounding amorphous region soft segments, which was reflected in the lower tensile strength and larger elongation at the break of BTPU.

### 3.4. Thermal Properties of BTPU

#### 3.4.1. Glass Transition Temperature (T_g_) and Melting Temperature (T_m_) of BTPU

Figure 6 illustrates the differential scanning calorimetry heating curves of the three kinds of BTPUs. The *T*_g_ values of the three BTPUs are below −36 °C and the *T*_m_ values of the three BTPUs are above 160 °C. The *T*_g_ and the *T*_m_ decreased with the increasing soft segment content because the *T*_g_ and *T*_m_ of BTPU depend on the macromolecular structure of the soft and hard segments and the ratio of the soft segment to the hard segment. The better the flexibility the molecular chain of the soft segment was, the lower the crystallinity and the easier the relative movement. With the increase of soft segment content, the flexibility of the molecular chains of the synthesized product increased. Furthermore, the segment could move at lower energy, which was reflected in the decrease of the *T*_g_. The molecular chain regularity and crystallization capability of BTPU decreased due to the increase of the soft segment content. Therefore, the *T*_m_ also decreased.

Figure 6 shows two melting peaks in the BTPU samples with 60 wt% and 70 wt% BPD contents, which were probably caused by the presence of two crystal forms in the hard segment. Given the change of soft segment contents, some changes in the peak shape and peak value may have occurred. With the decreasing soft segment ratio, the peak value of BTPU increased. This case may be due to the decrease in the soft segment content, the corresponding increase in the hard segment content and a relative increase in the interaction of the hard segments.

#### 3.4.2. Thermal Stability of BTPU

Figure 7 shows the thermal gravimetric analysis (TGA)/differential thermal gravimetric (DTG) analysis curves of the BTPU sample with 70 wt% BPD contents. The results indicated that the synthesized BTPU had typical decomposition features of carbon chain polymeric materials. The initial decomposition temperature of the BTPU was 323 °C, which is higher than those of some traditional TPUs [37,38] with an initial decomposition temperature below 300 °C, demonstrating that the novel BTPU had good thermal stability. The weight loss of the BTPU sample was low when the temperature was less than 323 °C due to low water and some molecular residues in the sample. The TGA curve shows that the BTPU had a higher weight loss rate within 323 °C to 500 °C and the final solid residue was only 4.03%.

As shown in Figure 7, two peaks are in the DTG curve at 352 °C and 426 °C. This indicated that the BTPU macromolecule consists of two kinds of molecular structures: The soft segment of BPD and the hard segment synthesized from BDO and MDI. This finding further confirmed that the structure of BTPU is of the alternating block type. The low-temperature decomposition peak in the DTG curve was mainly caused by the decomposition of the soft segment of BPD. However, the soft segment and hard segment decompositions were not isolated processes because the free radicals and energy produced during the decomposition of soft segments of BPD activated the decomposition of the hard segments.

### 3.5. Degradation Property of BTPU

As seen from the degradation loss curves of BTPU with different BPD contents in a lipase solution (Figure 8), BTPU-3 exhibited a weight loss ratio of 36.7% after 14 days in the lipase solution. Compared to some traditional TPU [39], which do not have significant degradation in a lipase solution, the novel BTPU had superior biodegradability. This can be explained, as the soft segment (BPD) comprising six kinds of aliphatic polyester repeat units and a carbon-carbon double bond had good flexibility and many ester bonds, so the macromolecular main chain of the BTPU was easily attacked and decomposed by lipase.

Meanwhile, the degradation loss rate of the BTPU in the lipase solution showed a distinct increasing trend with the increase of the degradation time, and the degradation loss rate of BTPU increased with the increase of the BPD content. The hard segment was the major part responsible for crystalline formation in the BTPU. The increased BPD content decreased the content of the hard segment, resulting in the increased polymer molecular chain flexibility and decreased BTPU crystallinity. Due to the decrease of the physical crosslinking points formed by the hard segment, the lipase easily penetrated the molecule interior. This condition resulted in a substantial increase in the degradation degree.

## 4. Conclusions

A novel BTPU with controllable biodegradability, good thermal properties, and mechanical properties can be successfully fabricated. First, the more flexible BPD (soft segment) was synthesized by using the melt polycondensation of five bio-based aliphatic monomers (SU, SA, FA, PDO, and BDO). Second, excessive MDI was reacted with the BPD, which led to a new functional group of -NHCO- at the end of BPD macromolecules. Finally, the BDO as the extender reacted with the functional group of -NHCO- in the BPD, whereas the BDO also reacted with excess MDI to form the hard segment of the BTPU. The BTPU containing 60 wt% of BPD content with a *T*_g_ of approximately −36 °C and a *T*_m_ of about 200 °C had a tensile strength of about 30 MPa, whereas its elongation at break exceeded 800%. With the soft segment decreasing, the tensile strength of the BTPU increased and the elongation at break of the BTPU decreased. The BTPU also had a superior degradation rate in lipase solution and the degradation rate increased with rising BPD content.

## Figures and Tables

**Figure 1 polymers-11-01121-f001:**
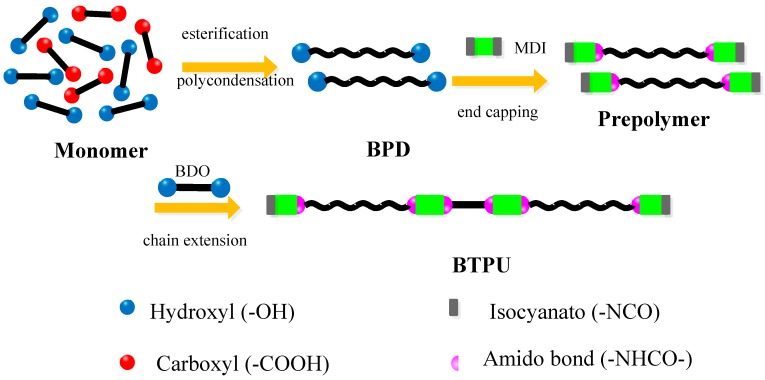
Schematic illustration of biodegradable thermoplastic polyurethane (BTPU) formation.

**Figure 2 polymers-11-01121-f002:**
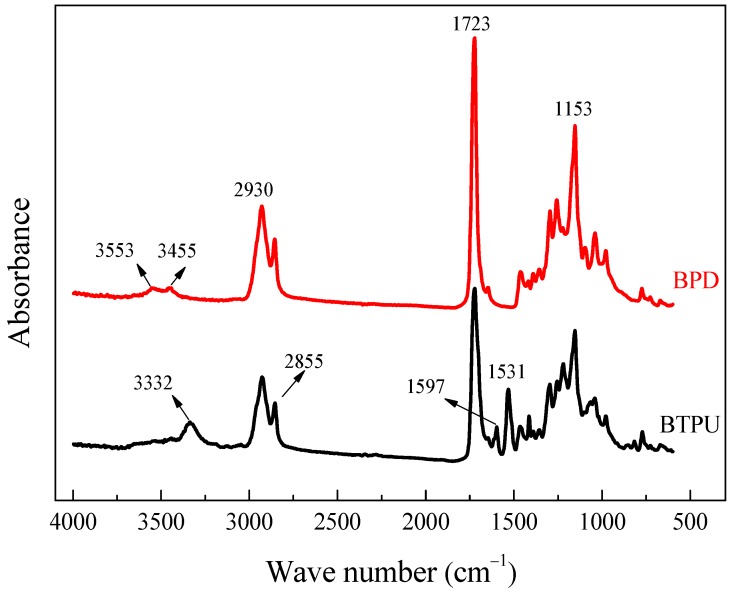
FTIR spectra of biodegradable polyester diols (BPD) and BTPU.

**Figure 3 polymers-11-01121-f003:**
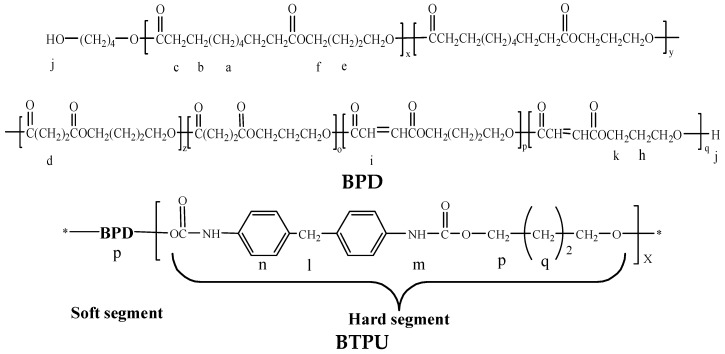
Molecular structural formulas of BPD and BTPU.

**Figure 4 polymers-11-01121-f004:**
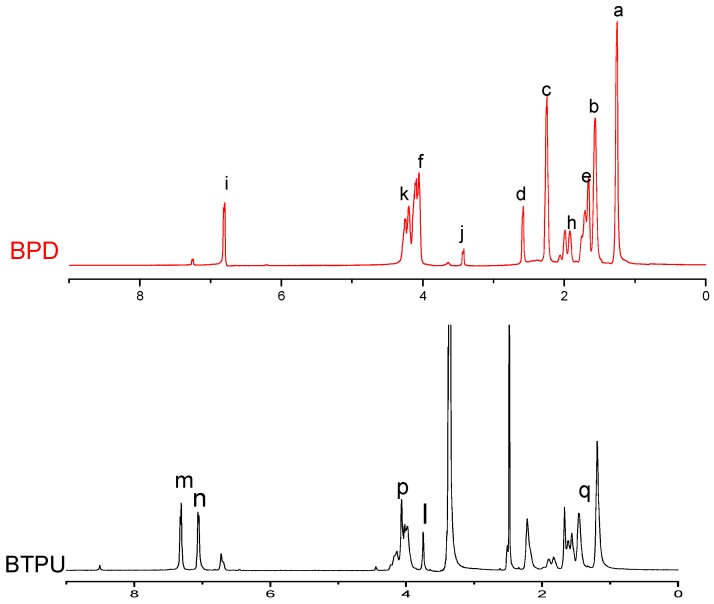
^1^H-NMR spectra of BPD and BTPU.

**Figure 5 polymers-11-01121-f005:**
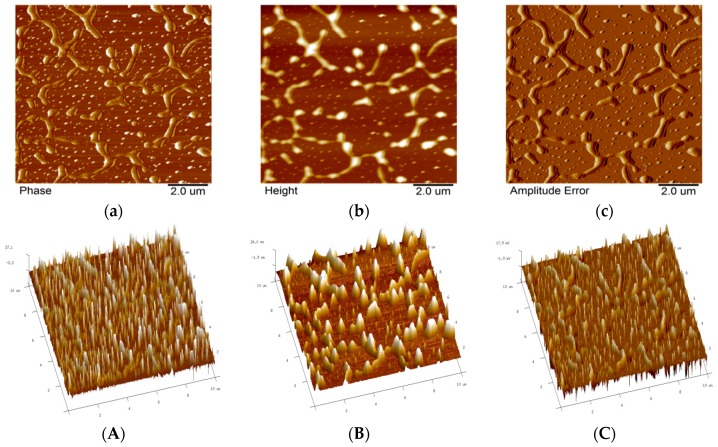
Atomic force microscopic images of BTPU. (**a**) 2-D phase image, (**b**) 2-D height image, (**c**) 2-D amplitude error image. (**A**) 3-D phase image, (**B**) 3-D height image, and (**C**) 3-D amplitude error image.

**Figure 6 polymers-11-01121-f006:**
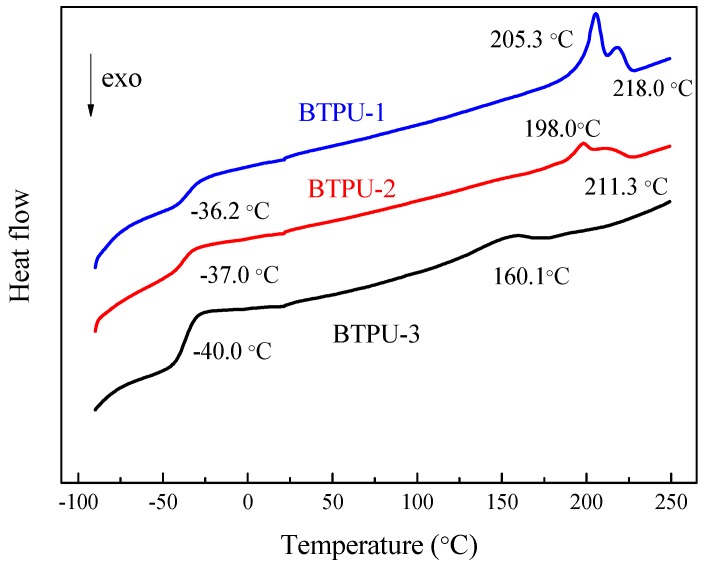
Thermal properties of BTPU with different BPD contents.

**Figure 7 polymers-11-01121-f007:**
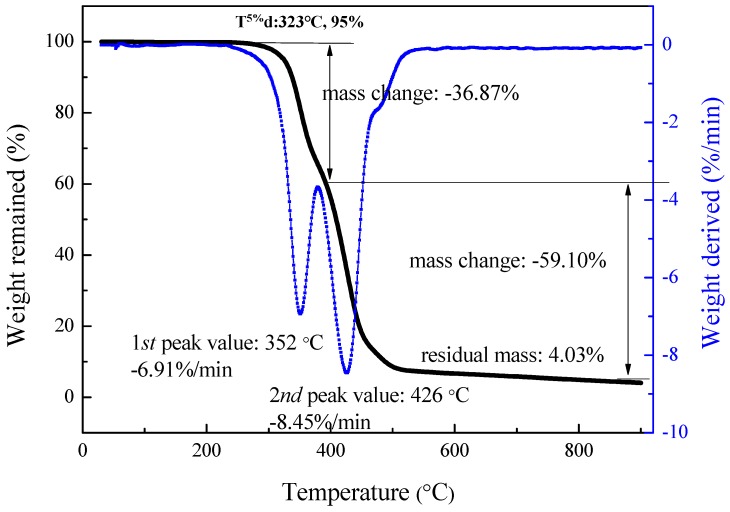
Thermal gravimetric analysis/differential thermal gravimetric analysis curves of BTPU sample.

**Figure 8 polymers-11-01121-f008:**
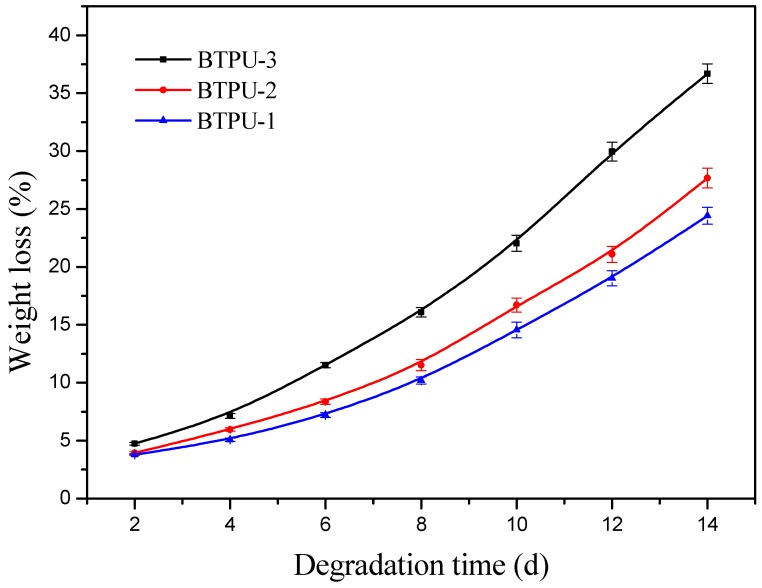
Weight loss of BTPU with different BPD contents after enzymatic degradation.

**Table 1 polymers-11-01121-t001:** Compositions of various biodegradable thermoplastic polyurethanes (BTPUs).

Sample Name	Soft Segment (BPD) Content (wt%)	Hard Segment Content (wt%)	
		MDI	BDO
BTPU-1	60	30.45	9.55
BTPU-2	70	23.27	6.73
BTPU-3	80	16.09	3.91

Note: NCO/OH = 1.

**Table 2 polymers-11-01121-t002:** Molecular weight and molecular weight distribution of BTPU.

Sample Name	Soft Segment (BPD) CONTENT (wt%)	$\bar{M_{n}}$	$\bar{M_{w}}$	$\bar{M_{w}}/\bar{M_{n}}$
BTPU-1	60	40,004	90,809	2.27
BTPU-2	70	51,236	167,541	3.27
BTPU-3	80	63,545	240,200	3.78

Note: $\bar{M_{n}}$ of the BPD in Table 2 is 3825.

**Table 3 polymers-11-01121-t003:** Some mechanical properties of various BTPUs.

Sample Name	Soft Segment (wt%)	Tensile Strength (MPa)	Elongation at Break (%)	Hardness (A)
BTPU-1	60	30.2 ± 1.5	802 ± 15	87 ± 1
BTPU-2	70	21.3 ± 1.2	1008 ± 17	80 ± 1
BTPU-3	80	13.5 ± 1.2	1405 ± 20	60 ± 1

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
