# Peer review of "Fabrication and Properties of a Bio-Based Biodegradable Thermoplastic Polyurethane Elastomer"

_polymers, 2019, doi:10.3390/polym11071121_

Round 1
Reviewer 1 Report
The authors presented an interesting topic on TPU and tuned the properties of TPU by manipulating the preparation method. However, there are similar study (both synthesis and characterization) in the past, what are the novelty of this manuscript? It is recommended that the authors provide more background information and relevant references. Besides, the manuscript needs language proofreading. Many of the sentences are long and redundant, which affected the readability. The authors also need to pay attention to the reference style/format in the manuscript.
The following are the detailed comments:
1. Line 11, the author mentioned “”the more flexible…”, what does it compare to? The same goes to Line 304.
2. Line 18, “superior biodegradability…”, similar to the last comment, what is the benchmark? The same goes to Line 294.
3. Line 27-28, “…chemical corrosion resistance, and biocompatibility.” What are the reference?
4. Line 28, “TPU is ……industry and agriculture.” Could the author elaborate or more specific about the field?
5. Line 30-31, what is the logic of the sentence? Whether a polymer leads to more or less environmental problem is not determined solely by its biodegradability, one should look at its life cycle. What does the “vigorous development” mean?
6. Line 34-35, “biomass resources……human being in the future.” Many hold the opinion that the biomass resources used for polymer synthesis should not compete with the food supply for human being. Therefore, they will not recommend to use the biomass resources for polymer.
7. Line 42-49, are “PLA-based polyesterurethanes” and “rigid polyurethane” mentioned in the references the same as above so-called thermoplastic polyurethane (TPU)?
8. Line 47-48, “the rigid …about 50 wt% and 60 wt% in …after 12 weeks…. processability”, it is hard to understand.
9. The “Section 2.1. Materials” could be more concise by summarizing in a Table.
10. Line 87, “brought” ?
11. Line 90-94, please rephrase the sentence. It is hard to follow.
12. Line 107-108, how BTPU 1-3 was made separately? Are they all produced at a temperature range at 100-120C?
13. The “Section 2.4”, could the author elaborate more on the purpose of the test and the experimental procedure including critical testing parameters? Not all people has access to the test standard.
14. Line 274, “BTPU” has a good thermal stability.” was the result compared to the commercial product or petroleum-based TPU? How is the conclusion made?
15. Line 279, “…subsequent processing…”, this is a meaningless statement without any specific condition.
Author Response
Response to Reviewer 1 Comments
Dear Reviewer:
We appreciate reviewer very much for your positive and constructive comments and suggestions on our manuscript entitled “Fabrication and properties of a bio-based biodegradable thermoplastic polyurethane elastomer”. (ID: polymers-497021).
On the basis of your recommendation and comments, we have revised the manuscript, and the followings are our responses to the reviewer.
Comment of reviewer #1
The authors presented an interesting topic on TPU and tuned the properties of TPU by manipulating the preparation method. However, there are similar study (both synthesis and characterization) in the past, what are the novelty of this manuscript? It is recommended that the authors provide more background information and relevant references. Besides, the manuscript needs language proofreading. Many of the sentences are long and redundant, which affected the readability. The authors also need to pay attention to the reference style/format in the manuscript.
Answer to reviewer #1.
The novelty of this manuscript is to prepare the novel bio-based biodegradable TPU, in which the soft segment consists of at least 6 repeat polyester units and has more flexibility.
As suggested as reviewer #1, we have revised the manuscript, especially in the revision of references, language proofreading. And we answer the detailed comments as follows.
Point 1: Line 11, the author mentioned “”the more flexible…”, what does it compare to? The same goes to Line 304.
Response to Point 1: Polymeric chains with -CH2-CH=CH-CH2- are more flexible than -CH2-CH2-CH2-CH2-. “-CH2-CH=CH-CH2-” has lower internal rotational barrier than “-CH2-CH2-CH2-CH2-”.
The reference [33] named as “Effect of double bonds on the dynamics of hydrocarbon chains” shows that a double bond introduced into a hydrocarbon chain increases the global relaxation rates with respect to the corresponding saturated chain, which increase the molecular chain flexibility.
Point 2: Line 18, “superior biodegradability…”, similar to the last comment, what is the benchmark? The same goes to Line 298.
Response to Point 2: As is well know that poly(lactic acid) is a biodegradable plastic. Being biodegraded with the same degradation conditions described in this paper, weight loss of the poly(lactic acid) is only about 1% within 14 days. So, the BTPU with weight loss of about 35% in this paper has superior biodegradability.
Point 3: Line 27-28, “…chemical corrosion resistance, and biocompatibility.” What are the reference?
Response to Point 3: As suggested by the reviewer, We have added 2 references for the “biocompatibility”, meanwhile, we deleted the “chemical corrosion resistance ” in revised manuscript.
Point 4 : Line 28, “TPU is ……industry and agriculture.” Could the author elaborate or more specific about the field?
Response to Point 4: As suggested by the reviewer, we have revised the manuscript and elaborate more specific about the field.
Point 5: Line 30-31, what is the logic of the sentence? Whether a polymer leads to more or less environmental problem is not determined solely by its biodegradability, one should look at its life cycle. What does the “vigorous development” mean?
Response to Point 5: As suggested by the reviewer, we have revised the manuscript and deleted the sentence of “inducing environmental pollution problems with the vigorous development of TPU. Thus,”. Then the sentence has correct logic.
Point 6: Line 34-35, “biomass resources……human being in the future.” Many hold the opinion that the biomass resources used for polymer synthesis should not compete with the food supply for human being. Therefore, they will not recommend to use the biomass resources for polymer.
Response to Point 6: Yes, we are in agreement with the opinion the reviewer pointed out, and people should not apply the biomass resources for food supply to synthesize the polymer. However, some biomass resources, such as straw and other agricultural products processing wastes, also can be used to prepare the bio-based monomer.
Point 7: Line 42-49, are “PLA-based polyesterurethanes” and “rigid polyurethane” mentioned in the references the same as above so-called thermoplastic polyurethane (TPU)?
Response to Point 7: Yes, “PLA-based polyesterurethanes” and “rigid polyurethane” are the same as above so-called thermoplastic polyurethane.
Point 8: Line 47-48, “the rigid …about 50 wt% and 60 wt% in …after 12 weeks…. processability”, it is hard to understand.
Response to Point 8: As Reviewer suggested, we have re-written this part according to the Reviewer’s suggestion in revised manuscript.
Point 9: The “Section 2.1. Materials” could be more concise by summarizing in a Table.
Response to Point 9: We are in agreement with the suggestion pointed out by the reviewer, but the “materials” is not been listed as a table in the journal Polymers, so we do not revise this section.
Point 10: Line 87, “brought” ?
Response to Point 10: We have replaced the “brought” with “bought” in revised manuscript.
Point 11: Line 90-94, please rephrase the sentence. It is hard to follow.
Response to Point 11: As the reviewer suggested, we have revised the sentences.
Point 12: Line 107-108, how BTPU 1-3 was made separately? Are they all produced at a temperature range at 100-120℃?
Response to Point 12: BTPU-1, BTPU-2, and BTPU-3 were made separately by using different weight ratio of Soft segment content to Hard segment content listed in Table 1. BTPU 1-3 were all produced at a temperature range at 100-120℃.
Point 13: The “Section 2.4”, could the author elaborate more on the purpose of the test and the experimental procedure including critical testing parameters? Not all people has access to the test standard.
Response to Point 13: As the reviewer suggested, in the revised manuscript, we have elaborated more on the purpose and the experimental procedure including critical testing parameters.
Point 14: Line 274, “BTPU” has a good thermal stability.” was the result compared to the commercial product or petroleum-based TPU? How is the conclusion made?
Response to Point 14:. The Td of the BTPU is about 320℃, which is higher than the common polymer processing temperature (below 300 °C). Then we conclude that the BTPU has good thermal stability.
Point 15: Line 279, “…subsequent processing…”, this is a meaningless statement without any specific condition.
Response to Point 15: As suggested by the reviewer, we have rephrased this sentence in revised manuscript.
Again, the authors appreciate your time, recommendation, and construction comments.
Thank you so much.
Sincerely yours
Qingguo Wang

Reviewer 2 Report
Work is good, but manuscript is written very badly. Manuscript follows the pattern of "did this, observed that". No scientific explanation anywhere in the manuscript especially in biodegradation studies. Overall good work but mediocre manuscript.
Author Response
Response to Reviewer 2 Comments
Dear Reviewer:
We appreciate reviewer very much for your positive and constructive comments and suggestions on our manuscript entitled “Fabrication and properties of a bio-based biodegradable thermoplastic polyurethane elastomer”. (ID: polymers-497021).
On the basis of your recommendation and comments, we have revised the manuscript, and the followings are our responses to the reviewer.
Comments of reviewer #2
Work is good, but manuscript is written very badly. Manuscript follows the pattern of "did this, observed that". No scientific explanation anywhere in the manuscript especially in biodegradation studies. Overall good work but mediocre manuscript.
Response to reviewer #2:
As suggested by Reviewer #2, in revised manuscript, we have rewrote some sentences which are hard to understand.
Again, the authors appreciate your time, recommendation, and construction comments.
Thank you so much.
Sincerely yours
Qingguo Wang

Reviewer 3 Report
This paper introduces the preparation of a new biodegradable thermoplastic elastomer, with detailed analysis of its content elements, molecular structural formulas, micromorphological structure, mechanical properties, thermal properties, and degradation property. This paper is well written, with detailed experimental procedures and results clearly presented. A few comments to the authors before this paper can be accepted and published.
1. The title should be “Preparation and properties of a biodegradable thermoplastic polyurethane elastomer” I don’t think the elastomer is bio-based as based on the description from the paper, it is a synthetic chemical but not made though bio-methods.
2. Page 4, Section 2.4.3: is this section the experimental method for Section 3.1.3? It’s kind of difficult to relate these two sections.
3. Page 4, Sections 2.4.5 and 2.4.6: The order of these two sections should be switched as their experimental results sections in Section 3 are in a reversed order. Also, which county’s national standards are ISO 37-2011 and ISO 7619-1-2010?
4. Page 5, Line 178: Section “3.1.21” should be “3.1.2”.
5. Page 8, Line 247: What are “Tg” and “Tm”?
6. Page 9, Fig. 7: Please make the solid line of “weight remained” bold as to not be hidden by the “weight derived” dash-line.
Author Response
Response to Reviewer 3 Comments
Dear Reviewer:
We appreciate reviewer very much for your positive and constructive comments and suggestions on our manuscript entitled “Fabrication and properties of a bio-based biodegradable thermoplastic polyurethane elastomer”. (ID: polymers-497021).
On the basis of your recommendation and comments, we have revised the manuscript, and the followings are our responses to the reviewer.
Comments of reviewer #3
This paper introduces the preparation of a new biodegradable thermoplastic elastomer, with detailed analysis of its content elements, molecular structural formulas, micromorphological structure, mechanical properties, thermal properties, and degradation property. This paper is well written, with detailed experimental procedures and results clearly presented. A few comments to the authors before this paper can be accepted and published.
Response to comments of reviewer #3:
We have revised the manuscript according the comments of reviewer #3, and we answer the detailed comments as follows.
Point 1: The title should be “Preparation and properties of a biodegradable thermoplastic polyurethane elastomer” I don’t think the elastomer is bio-based as based on the description from the paper, it is a synthetic chemical but not made though bio-methods.
Response to Point 1: In this paper, we bought the bio-based monomers materials from some Biological Technology Co., Ltd. Unfortunately, we did not verify them by measuring the 14C element of the monomers.
Point 2: Page 4, Section 2.4.3: is this section the experimental method for Section 3.1.3? It’s kind of difficult to relate these two sections.
Response to Point 2: Yes, Section 2.4.3 introduces the experimental method for GPC, which supplies the Mn and Mw for the results and discussions of Section 3.1.3.
Point 3: Page 4, Sections 2.4.5 and 2.4.6: The order of these two sections should be switched as their experimental results sections in Section 3 are in a reversed order. Also, which county’s national standards are ISO 37-2011 and ISO 7619-1-2010?
Response to Point 3: As suggested by the reviewer, in revised manuscript, we have revised some description about the ISO in Section 2.4.5 and 2.4.6, meanwhile we also switched the order of Sections 2.4.5 and 2.4.6.
Point 4: Page 5, Line 178: Section “3.1.21” should be “3.1.2”.
Response to Point 4: As suggested by the reviewer, we have made correction in the manuscript.
Point 5: Page 8, Line 247: What are “Tg” and “Tm”?
Response to Point 5: “Tg” is the abbreviation of “Glass transition temperature” and “Tm”is the abbreviation of “Melting temperature”.
Wehaverevised the manuscript and given full names and abbreviations for the first time in the manuscript.
Point 6: Page 9, Fig. 7: Please make the solid line of “weight remained” bold as to not be hidden by the “weight derived” dash-line.
Response to Point 6: We have made correction according to the Reviewer’s comments.
Again, the authors appreciate your time, recommendation, and construction comments.
Thank you so much.
Sincerely yours
Qingguo Wang

Round 2
Reviewer 1 Report
Thanks for the authors’ reply!
The polymers journal is an internationally recognized prestigious journal and the manuscript should be carefully checked by an English editor rather than the reviewers. However, from the revision, it seems the manuscript still lacks the proofreading.
1. Although the authors made justification to the reviewers, the readers (not limited to the reviewers) may still be confused by the novelty of the manuscript. Here are some advices to the authors:
First of all, if this is a novel bio-based biodegradable TPU, the author should address it clearly to take the credit;
Secondly, with regard to “…more flexibility…”, the authors clearly showed that carbon-carbon double bonds are more flexible than C-C single bond. But in the context of this manuscript, the flexibility of this synthesized novel TPU should compare to other TPUs; if the other TPUs don’t have such a double bond, then the TPU from this manuscript has the advantage. Comparison should always be “apple to apple” instead of “apple to orange”;
Thirdly, the authors should clearly address those novelties in the manuscript since the readers may not have such information, which can be part of the literature review in the Introduction Section.
2. Point 2, similarly, with regard to the word “superior” from “…superior biodegradability…” , the readers want to see the why the novel TPU is superior . However, such information cannot be found in the manuscript. Besides, comparison should always be “apple to apple” instead of “apple to orange”, otherwise, it is meaningless;
3. Point 6, It is suggested that the authors change the wording and clearly say biomass residues (or similar words);
4. Point 8, even after the revision, the sentence doesn’t make sense from the grammar viewpoint. The authors should read carefully from the original article.
5. Point 14, comparison should always be “apple to apple” instead of “apple to orange”, otherwise, if novel TPU is worse than the commercially available TPU, then the novel TPU does not gain advantage in this regard;
6. Line 137, does the author mean “spinning” instead of “spining”?
7. Figure 7, does the author mean “quantity or mass” instead of “quality”?
Author Response
Dear Reviewer:
We appreciate you very much for your positive and constructive comments and suggestions on our manuscript entitled “Fabrication and properties of a bio-based biodegradable thermoplastic polyurethane elastomer”. (ID: polymers-497021).
On the basis of your recommendation and comments, we have revised the manuscript again, and the followings are our responses to your comments.
Point 1: Although the authors made justification to the reviewers, the readers (not limited to the reviewers) may still be confused by the novelty of the manuscript. Here are some advices to the authors:
First of all, if this is a novel bio-based biodegradable TPU, the author should address it clearly to take the credit;
Secondly, with regard to “…more flexibility…”, the authors clearly showed that carbon-carbon double bonds are more flexible than C-C single bond. But in the context of this manuscript, the flexibility of this synthesized novel TPU should compare to other TPUs; if the other TPUs don’t have such a double bond, then the TPU from this manuscript has the advantage. Comparison should always be “apple to apple” instead of “apple to orange”;
Thirdly, the authors should clearly address those novelties in the manuscript since the readers may not have such information, which can be part of the literature review in the Introduction Section.
Response to Point 1:
As suggested by the reviewer, we have revised the “Introduction” and some sections of “Results and Discussion” of manuscript to address the novelty of the BTPU clearly.
In the 4th paragraph of Introduction section, we have cited references of some TPUs which can not have good biodegradability and good elasticity simultaneously. Being different from those TPUs, the novel BTPU in this manuscript does have both good biodegradability and good elasticity.
Also, above traditional TPUs in “Introduction” section do not have carbon-carbon double bonds, and the elongations at break of those traditional TPUs (less than 300%) are lower than that of the novel BTPU (more than 800%, and up to 1300%) in this manuscript.
The revised last paragraph of “Introduction” section which address the novelty of this manuscript is as follows.
“Being different from above TPUs which can not have good biodegradability and good elasticity simultaneously, this paper, using some bio-based aliphatic monomers[32][33] and technologies of polyester esterification reaction and chain extension, designed and prepared a novel bio-based BTPU which does have both good biodegradability and good elasticity. The BTPU contains a novel soft segment of aliphatic polyester and a hard segment. Notably, compared to some soft segments with 1 to 3 polyester repeat units in some traditional TPUs [29][30][31], the novel soft segment of the novel BTPU comprises six kinds of aliphatic polyester repeat units and has good biodegradability and flexibility. Also, the fumaric acid is explored to synthesize the novel soft segment because the carbon-carbon double bond in molecular main chain has the effect on improving the molecular chain relaxation rates [34], leading to the more flexible soft segment. This paper also studied the effects of molecular structure, molecular weight and soft segment ratio on the biodegradability, mechanical property and thermal property of the BTPU. This work can provide new research insights for developing environment-friendly and functional biodegradable polymer materials.”
Point 2: Point 2, similarly, with regard to the word “superior” from “…superior biodegradability…” , the readers want to see the why the novel TPU is superior . However, such information cannot be found in the manuscript. Besides, comparison should always be “apple to apple” instead of “apple to orange”, otherwise, it is meaningless.
Response to Point 2: As suggested by the reviewer, we have revised the manuscript by introducing some references of some TPUs in the 4th paragraph of “Introduction” section and “3.5 Degradation property of BTPU” section in order to highlight the novelty of the BTPU prepared in this manuscript. Compared to those traditional TPUs with some shortcomings, the novel BTPU has superior biodegradability.
Point 3: Point 6, It is suggested that the authors change the wording and clearly say biomass residues (or similar words);
Response to Point 3: As suggested by the reviewer, we have revised the manuscript and introduced some literatures about the biomass residues for preparing some monomers.
Point 4 : Point 8, even after the revision, the sentence doesn’t make sense from the grammar viewpoint. The authors should read carefully from the original article.
Response to Point 4: As suggested by the reviewer, we have corrected the sentence as follows, “The mass loss ratio of the rigid PU-F reached around 40% and 50% in the enzymatic experiments after 12 weeks”.
Point 5: Point 14, comparison should always be “apple to apple” instead of “apple to orange”, otherwise, if novel TPU is worse than the commercially available TPU, then the novel TPU does not gain advantage in this regard;
Response to Point 5: As suggested by the reviewer, we have revised the manuscript and introduced some literatures about thermal stability of TPU in section “3.4.2 Thermal stability of BTPU ”. Compared to those TPUs, the novel BTPU in this manuscript has good thermal stability.
Point 6: Line 137, does the author mean “spinning” instead of “spining”?
Response to Point 6: Yes, we have replaced the “spining” with “spinning” in revised manuscript.
Point 7: Figure 7, does the author mean “quantity or mass” instead of “quality”?
Response to Point 7: Yes, we have replaced the word “quality” with “mass” in revised manuscript.
Again, the authors appreciate your time, recommendation, and construction comments.
Thank you so much.
Sincerely yours
Qingguo Wang

Round 3
Reviewer 1 Report
Again for the Point 4: The mass loss ratio of the rigid PU-F reached around 40% and 50% in the enzymatic experiments after 12 weeks”.
How does PU-F has two numbers of mass loss ratio? The numbers must be from two experiments. The author should read the reference carefully to deliver the exact meaning from other people's work.
Author Response
Dear Reviewer:
We appreciate you very much for your positive and constructive comments and suggestions on our manuscript entitled “Fabrication and properties of a bio-based biodegradable thermoplastic polyurethane elastomer”. (ID: polymers-497021).
On the basis of your recommendation and comments, we have revised the manuscript again, and the followings are our responses to your comments.
Comment of reviewer
Again for the Point 4: The mass loss ratio of the rigid PU-F reached around 40% and 50% in the enzymatic experiments after 12 weeks”.
How does PU-F has two numbers of mass loss ratio? The numbers must be from two experiments. The author should read the reference carefully to deliver the exact meaning from other people's work.
Answer to reviewer
After reading the reference carefully, we found that the previous sentence does not deliver the exact meaning of that work. We have corrected the sentence as follows, “After the 12-week enzymatic experiments, the mass of rigid PU-Fs lost by about 40% to 50%.”
Again, the authors appreciate your time, recommendation, and construction comments.
Thank you so much.
Sincerely yours
Qingguo Wang
